# Towards Explainable AI Planning as a Service

**Michael Cashmore[1], Anna Collins[1], Benjamin Krarup[1], Senka Krivic[1],**
**Daniele Magazzeni[1], David Smith**

[1]King's College London, United Kingdom, {firstname.surname}@kcl.ac.uk

## Abstract

Explainable AI is an important area of research within which Explainable Planning is an emerging topic. In this paper, we argue that Explainable Planning can be designed as a *service* – that is, as a wrapper around an existing planning system that utilises the existing planner to assist in answering contrastive questions. We introduce a prototype framework to facilitate this, along with some examples of how a planner can be used to address certain types of contrastive questions. We discuss the main advantages and limitations of such an approach and we identify open questions for Explainable Planning as a service that identify several possible research directions.

## 1 Introduction

Explainable Artificial Intelligence (XAI) is an emerging research area in AI, motivated by the need to engender trust in users by explaining to them why the AI is making a particular decision.

While the main focus of XAI has been in Machine Learning, recently there has been growing interest in Explainable Planning, as shown by many planning contributions at the IJCAI International Workshops on XAI (XAI 2017; 2018) and the successful first ICAPS Workshop on Explainable Planning (XAIP 2018). Since the initial ideas of Smith (2012), there has been significant effort, in particular on the topic of Human-Aware Planning and Model Reconciliation (Chakraborti et al. 2017; Zhang et al. 2017; Sreedharan, Chakraborti, and Kambhampati 2018). Other significant works include topics such as Explainable Agency (Langley et al. 2017), moral values (Lindner, Mattmüller, and Nebel 2018), insights from social science (Miller 2019), and preferred explanations (Sohrabi, Baier, and McIlraith 2011).

A roadmap for Explainable Planning (XAIP) was proposed by Fox, Long, and Magazzeni (2017). They discussed some types of user questions that should be addressed. In particular, one type of question is of the form "*Why did you do A rather than B?*". We refer to this type of question as a *contrastive question*. To answer this kind of question one must reason about the hypothetical alternative "B", which likely means constructing an alternative plan where "B" is used rather than "A". The hypothetical alternative would be a plan that is not better than the one found by the planner or a plan which is better than the original one. Providing such

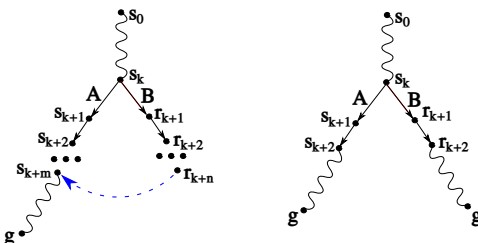

Figure 1: Generating contrastive explanations for a question "Why A rather than B?" at the state $s_k$.

a comparison between alternatives is what is called a *contrastive explanation*.

Fig. 1 shows an example of a contrastive question (Fox, Long, and Magazzeni 2017). Given a plan from the initial state $s_0$ to the goal $g$, the users might suggest an alternative action $B$ rather than $A$ at the state $s_k$. To provide a contrastive explanation means forcing action $B$ and then replanning from the resulting state to see the alternative plan. One possible alternative is a sequence of different actions that rejoin the original plan with a different cost (Fig. 1 left). Another possibility is a completely alternative sequence of actions that achieves the goal (Fig. 1 right). In both cases the user can compare costs to gain confidence on the quality of the plan found by the planner.

Given that planning is now used in safety-critical applications, for example oil-well drilling (Long 2018), explanations play a key role. Crucially, the greater the expense or risks in executing the plan, the more important the role of explanations for engendering trust in the users who are responsible and accountable for authorising the execution of a plan.

In this paper, we propose that Explainable Planning can be designed and constructed *as a service* – i.e., as a wrapper around an existing planning system that takes as input the current planning problem and domain model, the current plan, and the user's question. It must have the ability to invoke the existing planning system on hypothetical problems in order to address contrastive questions. This approach allows users to get explanations constructed from their own trusted planner and model. In complex or safety-critical domains this requirement is a crucial one. There is one im-

portant requirement, however; in order to effectively use the existing planning system, the XAIP service must be able to add constraints on the planning problem and domain model. However, the user will not accept an explanation generated using a model that differs from the original one that is potentially verified and trusted. Hence the explanation generated using the model revised with constraints has to be validated against the original model. In other words, the contrastive explanation should contain an executable plan which leads to the goal state that the original planner could have created using the original model.

Ideally, constraints over models should be described using a rich language designed for specifying constraints on the form of a desired plan. However, in many cases, these constraints can be compiled down into the domain model directly, which requires that the XAIP service have visibility and access into that model. This approach is otherwise agnostic about the domain model and the planner.

We have implemented this approach in a prototype framework for XAIP as a Service, in a PDDL setting, as it is a widely used planning language.

In particular, in this paper, we (1) present a prototype framework that enables Explainable Planning as a Service for contrastive questions, (2) describe some important categories of contrastive questions, (3) describe how the service compiles these contrastive questions into hypothetical planning problems that can then be solved by the existing planner to facilitate contrastive explanations, and (4) discuss the current state of our implementation and a roadmap for future work on Explainable Planning as a Service.

The paper is organised as follows: we start with a running example in the next section. In Section 3 we describe the XAIP as a Service framework, providing details for each component. In Section 4 we briefly discuss the framework that implements this approach. In Section 5 we discuss open issues and present a roadmap that identifies several possible research directions. Section 6 concludes the paper.

## 2   Running Example

As a running example throughout the paper, we use a simplified version of a safety-critical model that might be used in industry. The model describes a warehouse organisation delivery system. There are one or more robots that work together to move pallets from their delivery location to the correct storage shelf. Before the pallets can be stored the shelf must be scanned.

Fig. 2 defines the domain for this model. There are four temporal actions, *goto_waypoint*, *scan_shelf*, *load_pallet*, and *unload_pallet*. The *goto_waypoint* action is used for the robots to navigate the factory. It ensures that the shelf the robot is moving to is not occupied by another robot to stop congestion. The *scan_shelf* action is a sensing action. The *load_pallet* action loads the pallet from a shelf on to the robot. The robots do not have the ability to scan the shelf while holding a pallet. Finally, the *unload_pallet* action unloads the pallet onto a previously scanned shelf.

The problem we use as an example is shown in Fig. 3. There are two robots, two pallets, and six waypoints.

An example plan for this planning problem is shown in Fig. 4, its cost is its duration (20.003) which in this case is the optimal plan [1]. Upon close examination, the plan does not appear straight-forward. Both pallets are delivered by a single robot, and there are a lot of movements that appear to be inefficient. For example, the robot *Tom* moves away from waypoint *sh*6, even though there is an undelivered pallet at that location. It might seem more efficient to pick up that pallet while the robot is beside it. Without access to the domain and problem shown in Fig. 2 and 3, or without understanding of PDDL semantics, the behaviour of these robots will be opaque to a user, and explanation required.

```
(:types
  waypoint robot - locatable
  pallet
)
(:predicates
  (robot_at ?v - robot ?wp - waypoint)
  (connected ?from ?to - waypoint)
  (visited ?wp - waypoint)
  (not_occupied ?wp - waypoint)
  (scanned_shelf ?shelf - waypoint)
  (pallet_at ?p - pallet ?l - locatable)
  (not_holding_pallet ?v - robot)
)
(:functions
  (travel_time ?wp1 ?wp2 - waypoint))
(:durative-action goto_waypoint
  :parameters (?v - robot
    ?from ?to - waypoint)
  :duration(= ?duration
    (travel_time ?from ?to))
  :condition (and
    (at start (robot_at ?v ?from))
    (at start (not_occupied ?to))
    (over all (connected ?from ?to)))
  :effect (and
    (at start (not (not_occupied ?to)))
    (at end (not_occupied ?from))
    (at start (not (robot_at ?v ?from)))
    (at end (robot_at ?v ?to)))
)
(:durative-action scan_shelf
  :parameters (?v - robot
               ?shelf - waypoint)
  ...)
(:durative-action load_pallet
  :parameters (?v - robot ?p - pallet
               ?shelf - waypoint)
  ...)
(:durative-action unload_pallet
  :parameters (?v - robot ?p - pallet
               ?shelf - waypoint)
  ...)
```

Figure 2: A fragment of a robotics domain used as a running example. Some of the operator bodies have been omitted for space.

---

[1] The plan is obtained using the planner POPF (Coles et al. 2010). However our framework accounts for all PDDL2.1 planners.

```
(define (problem task)
(:domain warehouse_domain)
(:objects
    sh1 sh2 sh3 sh4 sh5 sh6 - waypoint
    p1 p2 - pallet
    Jerry Tom - robot
)
(:init
    (robot_at Jerry sh3)
    (robot_at Tom sh5)
    (not_holding_pallet Jerry)
    (not_holding_pallet Tom)
    (not_occupied sh1) (not_occupied sh2)
    (not_occupied sh4) (not_occupied sh6)
    (pallet_at p1 sh3) (pallet_at p2 sh6)
    (connected sh1 sh2) (connected sh2 sh1)
    (connected sh2 sh3) (connected sh3 sh2)
    ...
    (= (travel_time sh1 sh2) 4)
    (= (travel_time sh2 sh1) 4)
    (= (travel_time sh2 sh3) 8)
    (= (travel_time sh3 sh2) 8)
    ...
)
(:goal (and
    (pallet_at p1 sh6)
    (pallet_at p2 sh1))))
```

Figure 3: A fragment of the problem instance used in the running example.

```
0.000: (goto_waypoint Tom sh5 sh6) [3.000]
0.000: (load_pallet Jerry p1 sh3) [2.000]
2.000: (goto_waypoint Jerry sh3 sh4) [5.000]
3.001: (scan_shelf Tom sh6) [1.000]
4.001: (goto_waypoint Tom sh6 sh1) [4.000]
7.001: (goto_waypoint Jerry sh4 sh5) [1.000]
8.001: (scan_shelf Tom sh1) [1.000]
8.002: (goto_waypoint Jerry sh5 sh6) [3.000]
9.001: (goto_waypoint Tom sh1 sh2) [4.000]
11.002: (unload_pallet Jerry p1 sh6) [1.500]
12.503: (load_pallet Jerry p2 sh6) [2.000]
14.503: (goto_waypoint Jerry sh6 sh1) [4.000]
18.503: (unload_pallet Jerry p2 sh1) [1.500]
```

Figure 4: Plan generated from the example domain and problem with cost 20.003.

## 3 Providing explanations as a Service

We present an *XAIP Service* framework for providing contrastive explanations. Figure 5 summarises the approach taken by the framework, following these steps:

**Step 1:** The *XAIP Service* takes as input the planning problem and model, the plan, and the question from the user.

**Step 2:** The contrastive question implies a *hypothetical model* characterised as an additional set of constraints on the actions and timing of the original problem. These constraints can then be compiled into a revised domain model (HModel) suitable for use by the original planner.

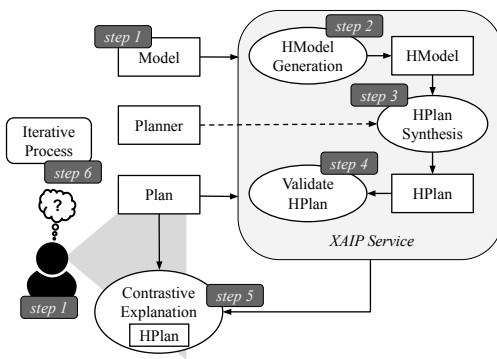

Figure 5: Architecture for Explainable Planning as a service, following the steps described in Section 3.

**Step 3:** The *original planner* uses the HModel as input to produce the *hypothetical plan* (HPlan).

**Step 4:** The XAIP Service validates the HPlan according to the original model.

**Step 5:** A contrastive explanation is constructed from the original plan and HPlan, and shown to the user.

**Step 6:** The user can choose to iterate the process from Step 1 with a new question. The user can choose to repeat the process with the original model and plan, or any HModel and HPlan.

The role of the HModel is to coerce the planner into creating the alternative plan that includes the actions and temporal constraints the user has in mind. One or more constraints must be added to the original model to create the HModel. We distinguish between three levels of abstraction in this process: In the first level the user question is given in natural language. The second level is a formal question that is derived from the natural language question. The formal question represents a set of constraints that are to be imposed upon the original model. Finally, the third level is a compilation of the formal question into the planning language. In our framework we focus on PDDL2.1 as the planning language, as we are interested in temporal and numeric planning problems. In this paper we describe the overall approach, and present a framework that encapsulates this process. The framework is modular, and allows different interfaces for providing user questions, and presenting explanations. The interface currently implemented in the framework allows the user to select a formal question directly. This represents constraints upon the plan. The set of formal question types from which the selection is made is described below in Section 3.1. For example, the user might have a question for the running example such as:

"At the point in the plan where action (*goto_waypoint Tom sh6 sh1*) is used, there's a pallet, so why doesn't Tom pick it up?"

From the user question above the following formal question is derived:

*Why is action A used in state S, rather than action B?*

Where action A is (*goto_waypoint Tom sh*6 *sh*1), action B is (*load_pallet Tom p*2 *sh*6), and the state S is the state in which action A was originally applied. This constraint enforces that the plan includes action B in state S instead.

Finally, this constraint can be compiled into the original model to produce the HModel, as described in Section 3.2.

## 3.1 Encoding User Questions

The user question is encoded as a set of constraints, which represent the formal question, and this is be done through an user interface where the user is guided to select the constraints that match their question. The questions we are interested in are *contrastive questions* of the form, *"Why A rather than B?"*, where A is the *fact* (i.e. what occurred in the plan) and B is the *foil* (i.e. the hypothetical alternative expected by the user). The formal questions currently handled by our approach are:

- "*Why is action A used in the plan, rather than not being used?*" This constraint would prevent the action A from being used in the plan.

- "*Why is action A not used in the plan, rather than being used?*" This constraint would enforce that the action A is applied at some point in the plan.

- "*Why is action A used, rather than action B?*" This constraint is a combination of the previous two, which enforces that the plan include action B and not action A.

- "*Why is action A used before/after action B (rather than after/before)?*" This constraint enforces that if action A is used, action B must appear earlier/later in the plan.

- "*Why is action A used outside of time window W, rather than only being allowed inside W?*" This constraint forces the planner to schedule action A within a specific time window.

One specific form of the question: "*Why is action A used, rather than action B?*", represented in Fig. 1, is "*Why is action A used in state S, rather than action B?*". This refinement forces the plan to include action B in state S, where B is an action (different from A) that is valid in that state. Using this constraint, the actions leading up to state S would remain unchanged, and the action A would still be allowed in other parts of the plan.

Deriving a formal question from a user question in natural language represents a significant research challenge. We discuss how a formal question might be identified automatically from natural language in Section 5.

While this set of formal questions covers a wide a range of possible situations and explanations, this does not comprise a complete set of constraints that could be applied to the problem. However, a single explanation does not exist in a vacuum, and a user might have a series of questions that will iteratively increase their understanding of the plan proposed by the planner, or the user can realise that the first question was not complete. For this reason, following ideas of Smith (2012), the framework allows the user to apply a sequence of formal questions to a single plan. For example, the general question "Why is action A used, rather than action B" can be seen as a combination of the first two questions,

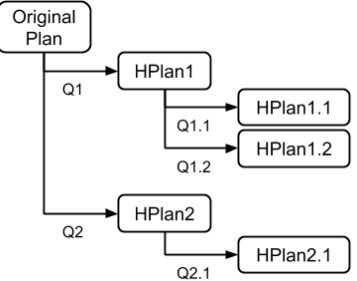

Figure 6: Example of iterative explanations.

which force action B to appear in the plan and not action A. This allows for a much wider range of explanations. Clearly the requirement of the plan being valid according to the original model must be satisfied at each iteration.

To allow the user to explore the space of hypothetical plans, it is possible to generate a tree of contrastive explanations in our framework, as illustrated in Fig. 6. Each node of the tree represents a hypothetical plan that was generated following a user question. The user can impose additional formal questions to a given plan in order to more precisely explore the behaviour that they are interested in. If the first question did not result in expected behaviour, the user can ask new questions to refine the hypothetical model until they reach a contrastive explanation that is satisfying to them.

As previously identified, the corresponding formal question in our running example is *"Why is action A used in state S , rather than action B?"*, where action A is (*goto_waypoint Tom sh*6 *sh*1), action B is (*load_pallet Tom p*2 *sh*6), and the state S is the state in which action A was originally applied.

## 3.2 Constructing the HModel

Constructing an HModel consists of taking the constraints in the formal question and compiling them into the planning domain model.

We give one example of this step, using the original model and plan shown in Section 2, and the question above. The compilation is formed such that the ground action *B* appears in the plan in place of the action *A*. Given a plan:

$$\pi = \langle a_1, a_2, \ldots, a_n \rangle$$

The ground action $a_i = A$ is replaced with *B*. The state directly after *B* has finished executing is found. This then becomes the new initial state $I'$ in the HModel. The framework uses the effects at each happening, computed by VAL (Howey, Long, and Fox 2004), up to the replacement action to compute $I'$.

In addition, the new initial state $I'$ is extended with a set of *timed-initial-literals* (TILs) which model the effects of actions that have started but not yet finished execution in the state selected by the user. A TIL is a tuple $\langle t, p \rangle$ where $p$ is the effect that is asserted, and $t$ is the time at which it is applied. Specifically, for each action $a_j$ not finished executing in state S, we add the TIL:

$$\langle start(a_j) + duration(a_j) - start(B), effect(a_j) \rangle$$

```
0.000: (goto_waypoint Tom sh5 sh6) [3.000]
0.000: (load_pallet Jerry p1 sh3) [2.000]
2.000: (goto_waypoint Jerry sh3 sh4) [5.000]
3.001: (scan_shelf Tom sh6) [1.000]
4.001: (load_pallet Tom p2 sh6) [2.000]
6.002: (unload_pallet Tom p2 sh6) [1.500]
7.502: (goto_waypoint Tom sh6 sh1) [4.000]
7.001: (goto_waypoint Jerry sh4 sh5) [1.000]
11.502: (scan_shelf Tom sh1) [1.000]
11.503: (goto_waypoint Jerry sh5 sh6) [3.000]
12.502: (goto_waypoint Tom sh1 sh2) [4.000]
14.503: (unload_pallet Jerry p1 sh6) [1.500]
16.004: (load_pallet Jerry p2 sh6) [2.000]
18.004: (goto_waypoint Jerry sh6 sh1) [4.000]
22.004: (unload_pallet Jerry p2 sh1) [1.500]
```

Figure 7: *HPlan* generated with a cost of 23.504. The replaced action is highlighted.

where $start(a_j)$ $(start(a_j))$ is the time at which the action $a_j$ $(B)$ started execution, $duration(a_j)$ is the planned duration of action $a_j$, and $effect(a_j)$ is the end effect of action $a_j$. In this example the action $(goto\_waypoint\, Jerry\, sh3\, sh4)$ is still executing in the state where our action is replaced. We add a TIL which makes $(not\_occupied\, sh3)$, and $(robot\_at\, Jerry\, sh4)$ true at time 2.999 in $I'$. This simulates finishing of the concurrent action execution.

A plan is then generated from this new state for the original goal, which gives us the plan:

$$\pi' = \langle a'_1, a'_2, \ldots, a'_n \rangle$$

The HPlan is then the initial actions of the original plan $\pi$ up to $a_i$, concatenated with the replaced action $B$ and the new plan $\pi'$:

$$\langle a_1, a_2, \ldots, a_{i-1}, B, a'_1, a'_2, \ldots, a'_n \rangle$$

The result is the HPlan shown in Fig. 7. The replaced action (B) is shown in bold. The initial actions before B are the actions from the original plan. The remaining actions are those of the new plan $\pi'$. As part of the service, the HPlan is validated with respect to the original model before a contrastive explanation is formed.

Note that in this HPlan, the user action is immediately reversed. This would not be a satisfactory explanation to the user, who wishes to see the plan in which carrying the pallet is essential to achieve the goal. Thus, it is not sufficient for the user suggested action to just be included in the plan, it must be part of the plan's key causal structure.

Fink and Yang (1992) define four categories of redundant actions, and Chrpa, Mccluskey, and Osborne (2012) present a simple algorithm to determine if an action is redundant in a sequential plan. As a post-processing step, after generating an HPlan, redundant actions of this kind can be detected. If suggested actions are not essential (redundant) to the HPlan, the planner needs to continue to search for additional plans until it finds one where the suggested actions are part of the causal structure for achieving the original goals. This additional search could potentially be made more efficient by introducing additional constraints (nogoods) into the HModel

that rule out alternative ways of achieving those goals, ultimately leaving only plans where the suggested actions are essential. For example, if the action B is redundant in the HPlan and action C is used instead of A, an additional constraint could be introduced to disallow C. In general, it is still an open question how to automatically infer useful nogood constraints from redundant HPlans.

An alternative is to allow the user to refine their question to rule out additional alternative solutions in which the suggested action is not essential. For example, the user's question might be expanded to include the constraint "and don't use actions C or D either". In either case it is clear that XAIP as a Service needs to follow an iterative approach where the planner generates a sequence of progressively more refined solutions, as additional constraints are imposed by nogoods, or by successive refinement of the user question. This is discussed further in Section 3.3.

Ideally, we would like to compile the original user question into an HModel that guarantees that suggested actions are essential to the causal structure of the plan (not redundant). However, it is an open question as to whether it is possible to do this. We suspect not. It is easy to force an action into a plan by adding a phantom effect to the action, and adding this phantom proposition to the goal. However, it is not obvious how to ensure that the action play an essential part in the achievement of other goals.

In addition to the example shown here, our framework includes compilations for all of the questions introduced in Section 3.1. Contrary to the compilation process of the presented example, the constraints posed from other questions are directly compiled into HModels which are used to produce HPlans.

### 3.3 Explainable Planning as an Iterative Process

Following the ideas of iterative processes by Smith (2012), we follow the same approach for explainable planning.

The user is able to use the framework to iterate the process by asking further questions, and refining the HModel. If the explanation does not completely satisfy the user, this allows the user to impose additional constraints that can be compiled into the HModel. For example, given the HPlan in Fig. 7, the user may have an additional question:

"Wait, shouldn't Tom have taken the pallet to its destination?"

Through the user interface with the framework the user selects a formal question. The formal question which encapsulates this user question is

*Why is the action A not used in the plan, rather than being used?*

Where action A is the action $(unload\_pallet\, Tom\, p2\, sh1)$. This constraint enforces that action A is used in the plan.

This constraint compiled into the HModel to enforce the user's suggestion, resulting in a new HModel. The *HModel* is then solved with the original planner to obtain the plan shown in 8.

```
0.000: (goto_waypoint Tom sh5 sh6) [3.000]
0.000: (load_pallet Jerry p1 sh3) [2.000]
2.000: (goto_waypoint Jerry sh3 sh4) [5.000]
3.001: (scan_shelf Tom sh6) [1.000]
3.002: (goto_waypoint Tom sh6 sh1) [4.000]
7.001: (goto_waypoint Jerry sh4 sh5) [1.000]
7.003: (scan_shelf Tom sh1) [1.000]
7.004: (goto_waypoint Tom sh1 sh6) [4.000]
11.004: (load_pallet Tom p2 sh6) [2.000]
13.004: (goto_waypoint Tom sh6 sh1) [4.000]
17.004: (unload_pallet Tom p2 sh1) [1.500]
17.005: (goto_waypoint Jerry sh5 sh6) [3.000]
20.005: (unload_pallet Jerry p1 sh6) [1.500]
```

Figure 8: *HPlan* generated with the second user constraint maintained, with a cost 21.505. The action suggested by the user is highlighted.

## 3.4 Forming Contrastive explanations

A contrastive explanation draws from the original plan $\pi$, the HPlan $\pi_H$ and the validation outcome. Defining a contrastive explanation is a complex task. In this paper we introduce a foundation for the contrastive comparison as a result of a comparison between the original plan and the HPlan.

The contrastive explanation can be defined as

$$CE = \langle comparison, Q \rangle$$

where *comparison* contains relevant information about the differences in plans that were caused by the user question $Q$:
$comparison(\pi, \pi_H) = \langle existing, removed, added, diffcost \rangle$
where:

- *existing* - actions in the plan which remained the same
- *removed* - actions which were removed from the plan
- *added* - actions which were added in HPlan and were not in original plan
- *diffcost* - the difference between the cost of the plans

By observing the *comparison*, a user can reason about the effect of the constraints that their question imposed and about the behaviour of the model. The validation outcome (performed against the original model) reassures the user about the validity of the HPlan. It also helps the user understand the difference between the costs of the two plans. The contrastive explanation for the HPlan in Fig. 8 is shown in Fig. 9.

## 4 XAIP as a Service Framework

For the purpose of evaluating the proposed approach to Explainable Planning as a Service we implemented a modular framework for domains and problems written in PDDL2.1. This prototype works with any planner capable of reasoning with PDDL2.1. The architecture of the framework is illustrated in Fig. 10. Interaction with a user is enabled through a console interface as well as a graphical user interface.

The *Controller* controls the behaviour of the XAIP framework and communicates with all other segments of the framework. The *XAIP Interface* creates a knowledge base

*existing*:

```
0.000: (goto_waypoint Tom sh5 sh6) [3.000]
0.000: (load_pallet Jerry p1 sh3) [2.000]
2.000: (goto_waypoint Jerry sh3 sh4) [5.000]
3.001: (scan_shelf Tom sh6) [1.000]
3.002: (goto_waypoint Tom sh6 sh1) [4.000]
7.001: (goto_waypoint Jerry sh4 sh5) [1.000]
7.003: (scan_shelf Tom sh1) [1.000]
17.005: (goto_waypoint Jerry sh5 sh6) [3.000]
20.005: (unload_pallet Jerry p1 sh6) [1.500]
```

*removed*:

```
9.001: (goto_waypoint Tom sh1 sh6) [4.000]
12.503: (load_pallet Jerry p2 sh6) [2.000]
14.503: (goto_waypoint Jerry sh6 sh1) [4.000]
18.503: (unload_pallet Jerry p2 sh1) [1.500]
```

*added*:

```
7.004: (goto_waypoint Tom sh1 sh6) [4.000]
11.004: (load_pallet Tom p2 sh6) [2.000]
13.004: (goto_waypoint Tom sh6 sh1) [4.000]
17.004: (unload_pallet Tom p2 sh1) [1.500]
```

$diffcost = 21.505 - 20.003 = 1.502$

Figure 9: The contrastive explanation that is presented to the user.

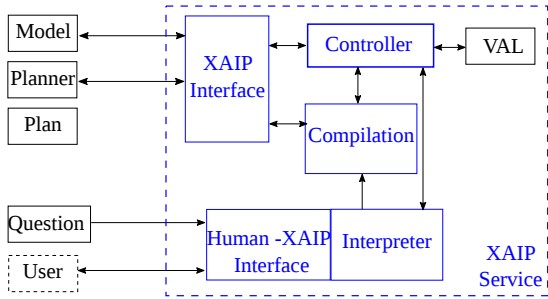

Figure 10: Architecture of the framework for Explainable Planning as a service.

from the PDDL files of the original model. This module also interacts with a planner. The *Compilation* module uses the constraints of the formal question to create the HModel. The validation module *VAL* (Howey, Long, and Fox 2004) is used as the validation technique.

The *Human-XAIP interface and interpreter* receives the questions from the user, creates a formal question instance, and demonstrates the explanation to the user. There are two implementations of this interface: a console and graphical user interface (Fig. 11). Both provide the same functionality, in which a user can see the plan, ask a formal question from the set of questions presented in Section 3.1, either by using a simple console interface or a set of forms for filling in the necessary details of the question. The output of the system is a visualisation of the original plan, HPlan, and VAL report. Actions in the plan are highlighted to correspond with the comparison described in Section 3.4.

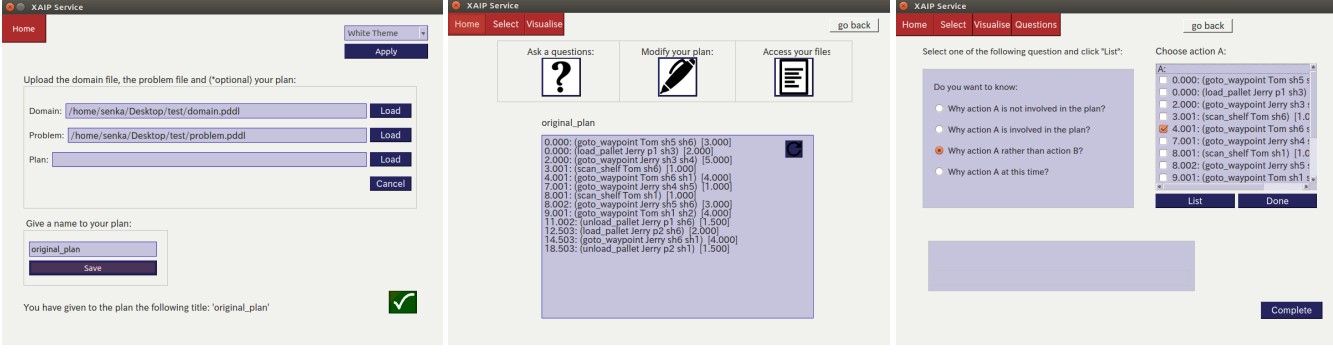

(a) Loading model        (b) Plan visualisation        (c) Question selection

Figure 11: Screenshots of the graphical user interface of the *XAIP Service* framework

## 5   Discussion

In order for Explainable Planning as a service to become effective, we discuss some challenges and outline some potential future work.

- *Understanding user questions.* The framework is modular to enable different ways for communicating a question to the AI system. Depending on the user interface a question could be given in different forms, for example speech, visual gestures or text input. Different technologies can be used to translate the question into a formal question such as: speech recognition, Natural Language Processing methods or human body tracking. Context can play a crucial role in understanding the question that the user asks. Borgo, Cashmore, and Magazzeni (2018) showed an example of a question requiring two different explanations depending on the context in which it was asked. In each case, improperly interpreting the question lead to an unsatisfying explanation. One promising direction for addressing this challenge is the use of *argumentation* (Cyras et al. 2019).

- *Formally categorising the set of questions that can be answered with contrastive explanations.* Although some philosophers, such as van Fraassen (1980), noted that "why"-questions can be implicitly or explicitly understood as: "why is A better than some alternative?", there might be questions in the planning space for which contrastive explanations are not well-suited. For example, if the user is simply trying to understand the conditions or requirements for various actions in the plan, or the causal or temporal structure of the plan, a contrastive explanation may not be appropriate. Instead it might be more appropriate to highlight the causal structure in an abstracted version of the plan. An important issue for future work is the development of a formal taxonomy of the types of questions that should be addressed using contrastive explanations (Miller 2018).

- *Expressing constraints on actions and plan structure.* A contrastive question requires creating a hypothetical planning model, which is often characterised by constraints on what actions are permissible in the plan and how they are arranged. For example, the question "Why did you use action A rather than action B for achieving P?" requires planning with the hypothetical model where B is required to be in the causal support for achieving P, but A is not in that causal support. This is substantially more difficult than just universally excluding A from the plan and forcing B into the plan because A or B might be required or prohibited elsewhere in the plan. Currently we do not have a good language for expressing these kinds of constraints. PDDL 3 allows the expression of simple constraints on the order in which goals are achieved, but does not have the ability to express constraints on action inclusion, exclusion, or ordering, and does not allow us to place more complex constraints on *how* something is achieved or on plan structure. We would like to be able to say something simple like "$\text{Supports}(B,P) \wedge \neg\text{Supports}(B,P)$". LTL will likely play a key role in defining the semantics of any such language, but additional concepts concerning plan structure are needed, such as the ability to specify that an action is part of the causal support for a goal or subgoal.

- *Compiling constraints into the HModel.* We showed examples of how a constraint derived from a user question could be compiled to form an HModel. However, providing compilations for more general constraints (like the one above) and ensuring their correctness is an important issue. Additionally, the compilation can lead to producing plans which might differ from the original plan in ways unrelated to the user question. We believe that the work on planning with preferences (Gerevini, Saetti, and Serina 2006) and state-trajectory constraints (Baier et al. 2009) is an important first step, but does not yet address the full range of constraints needed.

- *Forming and presenting contrastive explanations.* The form of contrastive explanation we provide, as discussed in Section 3.4 and shown in the GUI in Fig. 12, is a very simple one that presents the original plan and HPlan and highlights the action differences between them. Also, it is possible to obtain hierarchical contrastive explanations by asking consecutive questions. However, this does not show the causality of the plans, or the differences in their causal structure. Fig. 13 shows a possible composite causal representation for both the original plan and

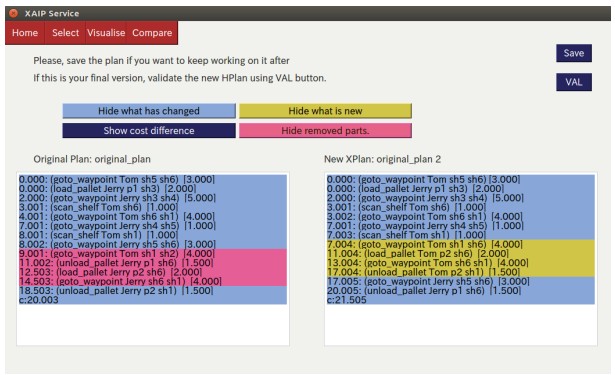

Figure 12: Output of the GUI in which differences in the plans are highlighted.

the HPlan, with the differences shown in different colors. This way of visualising the explanation can help to elucidate how the two plans achieve, or fail to achieve, the (sub)goals of the problem with respect to specific actions in the domain. However, for larger and more complex plans we expect that some form of abstraction will be necessary in order to effectively compare and contrast plans; the user might wants to see the important differences between two plans, not all the details. What counts as details remains an open research question, but is likely related to action costs and to the ease and importance of achieving various subgoals. Sreedharan, Srivastava, and Kambhampati (2018) have done some initial work in this area, and have considered milestones as important abstractions for purposes of explanation. The issues of what constitutes a good explanation, and how to visualize it or present it remain intertwined. Some synergy between researchers in planning, data visualization (e.g., Chakraborti et al. (2018) or Mennatallah et al. (2018)), and social sciences (Miller 2019) would be fruitful.

- *Providing explanations for complex questions.* In the presented approach, a user is able to iteratively ask questions to refine the explanation. If the explanation does not satisfy the user, or the question they have is more complex, this approach can provide the user with a deeper understanding. However, this process could be automated by analysing a more complex question the user might have, and decomposing it into several formal questions. In this case new constraints can be added to the HModel automatically until the explanation addresses the intended question and potentially the context it was asked within.

- *Assessing the effectiveness of explanations.* We believe it is crucial to be able to acquire evidence of the effectiveness of an explanation. In particular, if engendering trust is the motivation for Explainable Planning and XAI in general, then we should look at the actual experience of the users and check whether they gain confidence in the planner or not. For this, a vital step for planning researchers is to include *user studies* to assess the effectiveness of the explanations they are providing.

While of course this is not an exhaustive list of all the necessary next steps, it already provides an interesting set of challenges that should be addressed. Note that while we are advocating for Explainable Planning as a service, we are well aware that this is not the only way to provide explanations for planning. In particular, we envisage at least the following possible limitations that nevertheless represent important research questions that should be considered by the Explainable Planning community. First we acknowledge that contrastive explanations are not suitable to answer every type of question that the user might have. However, we argue that contrastive questions are common and that contrastive explanations therefore play a significant role, as also acknowledged by other researchers in Explainable AI, e.g, (Miller 2018). Second, by lifting the requirement that the explanation is generated by the planner used to generate the original plan, it could be possible to modify the search procedure used by the planner to generate explanations from a wider set of constraints. Third, we are assuming that there is no uncertainty in the original planning model and that the model is correct. However, this is not always the case and

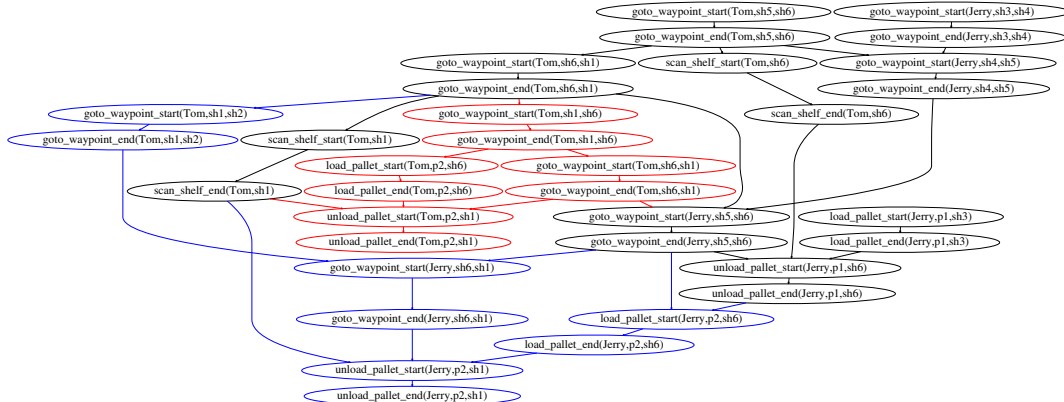

Figure 13: Example of a causal graph which demonstrates a comparison of the original plan and HPlan. Added actions are red, removed actions are blue.

for this, the body of research on model reconciliation plays a very important role (Chakraborti et al. 2017).

# 6 Conclusions

In this paper we have presented a prototype framework for Explainable Planning as a service, which we believe represents an effective way of providing explanations, particularly in safety critical domains. In such scenarios the user would not accept an explanation that is generated by a planner or a model different from the ones that they use and whose performance they trust. To this end, Explainable Planning as a service is based on providing explanations using the users' planners and models. Note that while the users can trust their planners and their models, there might still be reasonable questions on why a particular plan was found. This is where explanations are important, and we propose *contrastive explanations* to allow the user to compare the plan found by the planner with what the user was expecting.

The Explainable Planning as a Service framework is modular, and can be used with all planners and domains that ahere to PDDL2.1. In order to foster the use of Explainable Planning in robotics applications the proposed framework is now being integrated in ROSPlan (Cashmore et al. 2015).

We proposed a roadmap with some of the challenges that should be addressed for Explainable Planning to become effective, also highlighting promising directions. We believe that synergies between researchers in planning and in other disciplines, such as data visualization, social science, human-computer-interaction, and cognitive science, are key for the practical success of Explainable Planning.

**Acknowledgements**   This work was partially supported by Innovate UK grant 133549: *Intelligent Situational Awareness Platform*, and by EPSRC grant EP/R033722/1: *Trust in Human-Machine Partnerships*.

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
