# OpenReview forum: "Towards Explainable Planning as a Service"
_icaps-conference.org/ICAPS/2019/Workshop/XAIP — XAIP 2019_

### Official Review · AnonReviewer4 · 2019-05-13
**Interesting Approach/Framework outline paper for discussion**

**Rating:** 4
**Confidence:** 2

**Review:**

The paper follows up on one of the proposals by the Fox et al XAIP paper 2017, user questions "Why A rather than B?" with reference to a current plan. It thoroughly outlines and discusses the system that would need to be brought into place for answering this type of question based on the "as a service" idea meaning that the core of the answer generation is done by compilation into a planning proble, generating a new plan which serves as the basis for the answer.

The paper is mostly fairly high-level, discussing the various issues and possible problems and solutions in a lot of detail. I think this is perfectly adequate for presentation and discussion at the workshop. It certainly contains lots of thoughts that other workshop participants may take up upon/adapt to various other settings they may be interested in.

Two critical comments:

The new plan generated may differ from the previous plan in arbitrary ways unrelated to the user question. This is an inherent limitation of the approach, which should be discussed here. Maybe I overlooked that discussion; if not, I'd ask the authors to add such a discussion.

The title is very non-informative, and imho much too broad. Please make your headline more specific to the particular kind of questions, and answers, you are addressing here.

---

### Official Review · AnonReviewer1 · 2019-05-14
**Very good paper that perfect fits the workshop's aim**

**Rating:** 5
**Confidence:** 3

**Review:**


The paper presents the Explainable Planning (XAIP) framework  - introduced in the past IJCAI-XAI-17 workshop - to be provided "as a service". The key element of XAIP is the ability for the user to formulate contrastive questions (i.e., why did you do action A rather than B?). This would allow the user understanding the rationale behind the plan provided, transforming an opaque plan into a trustworthy one. To answer such a kind of question, the user has to (i) use its own planners; (ii) be able to add constraints on which the contrastive questions will be evaluated (aka hypothetical model, HModel); (iii) have a validation of the Hplan synthesised.

The approach is definitively interesting for the workshop, and for XAI broadly. The paper is well written and clear. The motivating example and the workflow are useful and gently  introduce to the reader the rationale behind XAIP.

COMMENTS:

-- In my view, the approach shown in Fig. 5 should be an iterative process, as the user might continue generating HModels, for example by asking to explain the behaviour of HModel^2 generated from HModel_1... Roughly, I consider XAIP an approach to better understand how a planner works in a given domain, so the trustworthy grows as the understanding grows. The authors also agree with this view, as they describe XAIPSaas as an iterative process in Sec 3.3. This might be also reflected into Fig.5

-- "The user question is encoded as a set of constraints, which represent the formal question, and this can be done through a dialogue where the user is guided to select the constraints that match their question." As far as I can understand from the paper, it seems there is a set of questions pre-defined (bullet points of Sec 3) - that the user can select from a GUI - where actions A and B act like a placeholder that the user can replace with domain variables. Then, each question is formalised using the approach of Smith, 2012.
If so - as it seems from Sec 5 -  I think using the term "dialogue" now is misleading as it seems the framework can encode naturale language into PDDL. To realise this, the authors might plan to employ word-embedding algorithms, that are able to represent word meaning into a N-dimensional vector space. Words with similar meaning are mapped to a similar position in the vector space. For example, “powerful” and “strong” are close to each other, whereas “powerful” and “Paris” are farther away. The word vector differences also carry meaning. For example, the word vectors can be used to answer analogy questions using simple vector algebra: “King” - “man” + “woman” ≈ “Queen” (Mikolov, Yih, and Zweig, 2013). This knowledge also grows over time, making the system able to "understand" different meaning of sentences on the base of the lexicon used.

-- An interesting feature that might be added to XAIP as a service would be the ability to perform analytics on the domain/problem. Roughly, it is a step-by-step approach that generate a HModel for each contrastive question. It might be usel to the user to select a "set" of contrastive questions that the system automatically encodes in a cascading HModels, then providing "analytics" about the behaviour of the system that summarises how the planner works in that domain as a whole in a "simulation" setting.

---

### Official Review · AnonReviewer5 · 2019-05-15
**Interesting paper on the challenge of explaining contrastive questions well**

**Rating:** 4
**Confidence:** 2

**Review:**

The idea of a wrapper around any planning system is a promising novel idea.

This sentence is slightly confusing: " In other words, the contrastive explanation should contain a valid plan that the original planner could have created using the original model." What does "valid" mean here? Does it just mean that the plan reaches the goal? Or is the main point that a contrastive explanation should be a realistic attempt by the planner, rather than say, easily outputting a plan that is clearly suboptimal?

It seems in some cases, there is no real reason one action is used before/after another, if both are optimal plans - even though this is the trivial case, perhaps it should be mentioned in the paper.

Section 3.2 is quite interesting, and clearly demonstrates some of the difficulty of answering contrastive questions informatively.

Overall this is a strong paper that highlights the difficulties of providing high-quality explanations to contrastive questions. This paper is highly relevant to the workshop.

minor comments:
	* "diffcost" in the tuple format in the second paragraph of 3.4 seems to be formatted as $diffcost$, I suggest using italics may be better

---

### Decision · Program_Chairs · 2019-05-15

**Decision:**

Accept

**Comment:**

The reviewers agree to accept. Please address all review criticism as best possible for the final paper version and its presentation at the workshop. Looking forward to discuss your work at the workshop!